# A Curriculum Batching Strategy for Automatic ICD Coding with Deep Multi-Label Classification Models

**DOI:** 10.3390/healthcare10122397

**Published:** 2022-11-29

**Authors:** Yaqiang Wang, Xu Han, Xuechao Hao, Tao Zhu, Hongping Shu

**Affiliations:** 1College of Software Engineering, Chengdu University of Information Technology, Chengdu 610225, China; 2Sichuan Key Laboratory of Software Automatic Generation and Intelligent Service, Chengdu University of Information Technology, Chengdu 610225, China; 3Department of Anesthesiology, West China Hospital, Sichuan University, Chengdu 621005, China; 4The Research Units of West China, Chinese Academy of Medical Sciences, West China Hospital, Sichuan University, Chengdu 621005, China

**Keywords:** automatic ICD coding, curriculum learning, minibatch gradient descent, deep learning

## Abstract

The International Classification of Diseases (ICD) has an important role in building applications for clinical medicine. Extremely large ICD coding label sets and imbalanced label distribution bring the problem of inconsistency between the local batch data distribution and the global training data distribution into the minibatch gradient descent (MBGD)-based training procedure for deep multi-label classification models for automatic ICD coding. The problem further leads to an overfitting issue. In order to improve the performance and generalization ability of the deep learning automatic ICD coding model, we proposed a simple and effective curriculum batching strategy in this paper for improving the MBGD-based training procedure. This strategy generates three batch sets offline through applying three predefined sampling algorithms. These batch sets satisfy a uniform data distribution, a shuffling data distribution and the original training data distribution, respectively, and the learning tasks corresponding to these batch sets range from simple to complex. Experiments show that, after replacing the original shuffling algorithm-based batching strategy with the proposed curriculum batching strategy, the performance of the three investigated deep multi-label classification models for automatic ICD coding all have dramatic improvements. At the same time, the models avoid the overfitting issue and all show better ability to learn the long-tailed label information. The performance is also better than a SOTA label set reconstruction model.

## 1. Introduction

The International Classification of Diseases (ICD) is a healthcare classification system initiated by the World Health Organization [1]. It is mainly used for uniformly coding diseases, symptoms, processes, injuries and other features contained in electronic medical records (EMRs) [2]. The coding results are widely used for epidemiological studies [3], the billing and reimbursement for medical services [4] and diagnostic information retrieval [5].

The ICD coding task requires professionals to master both medical knowledge and the ICD coding system. The coding process is time-consuming and laborious. In addition, different professionals may have different understandings of the complex ICD coding system. This may also lead to inconsistent coding results. Consequently, automatic ICD coding based on machine learning techniques has become an interesting research area in recent years [6,7].

Automatic ICD coding is usually regarded as a multi-label classification problem. The performance of automatic ICD coding is directly affected by extremely large label sets and imbalanced label distribution [8,9]. Recently, researchers have challenged these issues, mainly from two perspectives. One is reconstructing the label sets (ReLS) through designing different label-combining strategies [7,8,10]. The new label sets consist of multi-granularity labels. Another is building multi-level label hierarchical classifiers based on hierarchical joint learning mechanisms [9,11]. The main idea of these methods is to reduce the label space dimension and alleviate the imbalanced label distribution issue.

Deep multi-label classification models are widely used for automatic ICD coding, and the minibatch gradient descent (MBGD) is a widely used optimization algorithm for training the models [7]. In each iteration during a training procedure, the gradient updates of the model parameters are approximated, based on the examples in one batch. The studies show that a larger batch size results in a lower frequency of gradient updates and better gradient update approximation, but it may hurt generalization and result in finding poorer local optima [12].

In this paper, we further find that, during training, deep multi-label classification models for automatic ICD coding, which split the training data into batches with a simple but commonly used shuffling algorithm in MBGD, always result in the local data distribution of the batches (or the local batch data distribution) being inconsistent with the global training data distribution (see Figure 1). This phenomenon causes the local statistics of batches to vary substantially from the global statistics of the training data [13]. It will also hurt the model performance and generalizability (see Table 1).

To challenge the inconsistency problem, we propose a simple and effective curriculum batching strategy to conveniently replace the shuffling algorithm for splitting the training data into batches. Assuming that the uniform data distribution (UDD) is easier to be learned by deep multi-label classification models for automatic ICD coding than the shuffling data distribution (SDD), and the SDD is easier to be learned than the original imbalanced training data distribution (IDD), the proposed strategy applies three specific sampling algorithms to generate three batch sets with the three types of mentioned data distributions. The models then iterate each batch set, in turn, and learn the ICD coding experience from the easiest batch set to the hardest batch set.

A series of experimental results show that the proposed strategy can effectively avoid the overfitting issue and dramatically improve the automatic ICD coding performance compared with the three representative deep multi-label classification models and one state-of-the-art ReLS method for automatic ICD coding. Moreover, the improved models also have a better ability to learn the long-tailed label information. To the best of our knowledge, this is the first work to introduce a curriculum learning method for automatic ICD coding with deep multi-label classification models. In summary, the contribution of this paper is fourfold:This paper finds that the shuffling algorithm in the MBGD always causes the local data distribution of batches to be inconsistent with the global training data distribution, which hurts the model’s performance and generalizability;This paper alleviates the problems of poor generalization ability and low classification accuracy of the deep learning model in the field of ICD automatic coding;This paper greatly improves the learning ability of the automatic ICD coding model for data with long-tailed labels;This paper introduces the curriculum learning method into automatic ICD coding of deep multi-label classification model for the first time.

## 2. Related Work

### 2.1. Automatic ICD Coding

Automatic ICD coding is a widely discussed, hot research topic in the field of medicine and healthcare [6,7]. Many statistical machine learning methods have been applied to challenge this problem [6,14]. In recent years, with the widespread success of deep learning, it has also been introduced to solve the problem of automatic ICD coding [7,15]. The automatic ICD coding problem is formulated as a multi-label classification problem [16,17].

Extremely large label sets and imbalanced data distribution are two main issues directly impacting coding performance. Mullenbach presented a Convolutional Attention for Multilabel Classification (CAML) model incorporating an attention technique with convolutional neural networks [18]. Sadoughi extended this approach by adding a maximum pooling layer across all the multiple convolution layers [19]. To further improve the performance of CAML, Li integrating a multi-filter CNN architecture with a residual CNN architecture, which is called a multi-filter residual convolutional neural network [16]. Bhutto present a Deep Recurrent Convolutional Neural Network with Hybrid Pooling (DRCNN-HP), which considers the different lengths as well as the dependency of the ICD code-related text chunks [20]. Another current solution idea is to reduce the label space dimensions to alleviate the imbalanced data distribution. The label set reconstruction method [7,8,10] and the multi-label hierarchical classifier-based method [9,11] are two representative approaches. Differently from the previous studies, in this paper we explored further, from the perspective of how these problems affect the local model training procedure.

### 2.2. Mini-Batch Gradient Descent

Mini-batch gradient descent is the widely used optimization algorithm to train deep neural models [21,22], including the deep multi-label classification models for automatic ICD coding [7]. How to effectively use MBGD to train models is always a concern in machine learning and other related fields [23,24]. There are many aspects of research, including the effect of batch normalization [25], batch size and learning rate settings [12,26], etc. We studied the inconsistency problem between the local batch data distribution and the global training data distribution and attempted to avoid the risk of overfitting and to improve the model generalization ability.

### 2.3. Curriculum Learning

Elman first proposed a “starting small strategy” approach to learn a model with a curriculum [27]. Two methods were developed, including an incremental input method and an incremental memory method. The motivation behind these methods was that compound sentences make it a little difficult for neural networks to learn grammatical concepts in the early phases of training. Therefore, these methods make the models learn gradually, by varying the percentage of simple and compound sentences used during training or constraining the model’s capacity in the early phases of training [27].

Bengio et al. [28] revisited Elman’s approach and formalized the approach of starting training on easy examples first and then gradually introducing more complex examples during the training. This type of approach was named as “curriculum learning”, and it can achieve significant improvements in generalization. The core of curriculum learning is to develop a curriculum strategy to train the models from simple to complex.

Consequently, we proposed a simple and effective curriculum batching strategy for automatic ICD coding with deep multi-label classification models in order to solve the inconsistency problem mentioned earlier, improve the model generalization ability and avoid the risk of overfitting. The curriculum batching strategy generates and sequentially utilizes three types of batch sets composed of examples from easy-to-learn to hard-to-learn.

## 3. Methodology

In this section, we first review the original MBGD-based training procedure for deep multi-label classification models for automatic ICD coding. Then, we introduce the proposed curriculum batching strategy and how to integrate it conveniently into the MBGD-based training procedure, followed by the description of three specific sampling algorithms for obtaining batch sets satisfying the UDD, SDD and IDD.

### 3.1. MBGD-Based Training Procedure

The MBGD-based training procedure for deep multi-label classification models for automatic ICD coding is shown in Figure 2. This procedure consists of two modules: the data batch process and model training. In module one, the training data set D is divided into several mini-batches $\left(b_{1},b_{2},\ldots ,b_{K-1},b_{K}\right)$, which constitute the batch set ℬ. Second, each mini-batch b is sent to module two in turn. Every example (x, y) in mini-batch b contains an electronic case x and its corresponding ICD codes y, a real example (x, y) is shown in Figure 3. Third, the examples are sent to the model training module. The deep multi-label ICD coding model uses the electronic case x to predict its ICD codes and compares the predicted output with the real output y. Finally, module 2 calculates the losses for each sample and updates the model. The procedure is repeated until the model converges.

To be specific, there are six data or parameter processes and/or computational steps in the procedure, including:Batching Strategy: In this data process, a shuffling algorithm is commonly applied [26]; firstly, each example (x, y) in training data D is randomly sorted to generate a list. Then, the batch set $ℬ\left(b_{1},b_{2},\ldots ,b_{K-1},b_{K}\right)$ is formed by scanning the list with a sliding window with the size of M, i.e., the batch size. Thus, K, the number of batches b in ℬ, equals ⌈|D|/M⌉. This data process sometimes may be performed more than once;Parameter Initialization: In this parameter process, $\theta_{0}$ are usually drawn randomly from a distribution (e.g., uniform) as the initialization parameters;Loss Computation: Many kinds of deep multi-label classification models for automatic ICD coding can be applied in this step, including TextCNN, TextRNN and TextRCNN, which are compared in this paper. We uniformly note them as $f_{\theta_{k}}\left(x_{i}\right)$. $f_{\theta_{k}}\left(x_{i}\right)$ means that the model $f\left(x_{i}\right)$ takes the example $x_{i}$ as the input with the kth iteration’s parameter $\theta_{k}$. The loss of the current kth iteration, $ℒ\left(\theta_{k}\right)$, is the mean of the loss values obtained based on examples in ${ℬ}_{k}$. A loss value obtained based on one example $\left(x_{i},\;y_{i}\right)$ in ${ℬ}_{k}$ is noted as $L_{i}\left(y_{i},f_{\theta_{k}}\left(x_{i}\right)\right)$;Gradient Update Estimation: According to $\theta_{k}$, $f_{\theta_{k}}\left(x_{i}\right)$ and the examples in ${ℬ}_{k}$, the kth iteration’s gradient updates, $\Delta \theta_{k}$, can be estimated by the average of the gradients of the examples in ${ℬ}_{k}$, as shown in Figure 2;Parameter Update: The (k+1)th iteration’s parameter, $\theta_{k+1}$, is calculated, based on $\theta_{k}$ and $\Delta \theta_{k}$ with the hyperparameter learning rate η;Optimized Parameter Output: In general, the steps from 3 to 5 will loop E epochs until the model converges.

Theoretically, $\Delta \theta_{k}$ is the unbiased estimation of the kth iteration’s gradient updates. Consequently, a larger batch size leads to a better gradient update approximation and results in a lower frequency of gradient updating. However, there is no “free lunch”, in that it may also hurt generalization and result in finding poorer local optima [12].

Moreover, the commonly used batching strategy mentioned above makes each ${ℬ}_{k}$ carry different local statistics information from the global statistics information of ℬ, due to the inconsistency between the local data distribution of ${ℬ}_{k}$ and the global data distribution of ℬ. In addition, the extremely large label sets and imbalanced data distribution problems of automatic ICD coding also easily lead to a local overfitting on a small number of frequently occurring labels, due to the imbalanced label distribution of D. To challenge these problems, we propose a simple and effective curriculum batching strategy to conveniently replace the shuffling algorithm in the MBGD-based training procedure.

### 3.2. Curriculum-Batching Strategy

In the original MBGD-based training procedure for deep multi-label classification models for automatic ICD coding, ℬ is generated by the batching strategy introduced in Section 3.1, and the generated ℬ is used repeatedly by the following processes in the procedure and, sometimes, ℬ may be regenerated. To challenge the problems mentioned earlier, we propose a curriculum batching strategy (CBS) data batch process module in order to generate batch sets ${ℬ}_{UDD}$, ${ℬ}_{SDD}$ and ${ℬ}_{IDD}$ with three types of data distribution, i.e., UDD, SDD and IDD, respectively. The structure of the new module is shown in Figure 4.

The data batch process CBS can conveniently replace the original data batch process in Figure 2 (as depicted in Figure 4) except that the CBS will generate three batch sets, including ${ℬ}_{UDD}$, ${ℬ}_{SDD}$ and ${ℬ}_{IDD}$, which are sampled from the original training data, respectively, based on specific sampling algorithms. ${ℬ}_{UDD}$ is generated through a stratified sampling with replacement (SSR); ${ℬ}_{SDD}$ is generated based on the original shuffling algorithm; and ${ℬ}_{IDD}$ is generated by a probability sampling with replacement (PSR). The specific SSR and PSR methods will be described later; the original shuffling algorithm follows the batching strategy introduced in Section 3.1.

The intuitive assumption of the CBS is that ${ℬ}_{UDD}$ is easier to be learned by the deep multi-label ICD coding models than ${ℬ}_{SDD}$, and, in the same way, ${ℬ}_{SDD}$ is easier than ${ℬ}_{IDD}$. Moreover, ${ℬ}_{UDD}$ makes the models have an equally likely opportunity to learn the examples corresponding to each type of label, which can alleviate the imbalanced data distribution problem [29,30]. ${ℬ}_{SDD}$ allows models to learn every example corresponding to each type of label. ${ℬ}_{IDD}$ expects models to learn the ICD coding experience from examples through keeping the consistency between the local data distribution of ${ℬ}_{k}$ and the global data distribution of ℬ. So far, we have designed a curriculum strategy. According to the idea of curriculum learning [28,31], ${ℬ}_{UDD}$, ${ℬ}_{SDD}$ and ${ℬ}_{IDD}$ will be learned sequentially.

**SSR for generating**${ℬ}_{UDD}$: The labels of the ICD coding system are diverse, extremely large and imbalanced. These issues make the examples in D have unequal opportunities to be learned by the models, which easily lead to a local overfitting on a small number of frequently occurring labels in a fixed number of loops.

Stratified sampling has the ability to ensure that every characteristic of the population is properly represented in one sample [32]. Consequently, we designed a simple SSR to generate a batch set satisfying the requirement that every example corresponding to each type of label in D has an equal chance to be learned by the models.

First, SSR samples M labels randomly, where M equals the batch size. Then for each sampled label, SSR samples one example with a replacement randomly from the examples in D whose corresponding labels contain the sampled label. Finally, the M examples constitute one batch. The above procedure will be repeated N times, where N is the size of the batch set ${ℬ}_{UDD}$. The sampled N batches form the batch set ${ℬ}_{UDD}$. In this paper, N is uniformly set to ⌈|D|/M⌉.

**PSR for generating**${ℬ}_{IDD}$: Each ${ℬ}_{k}$ in ${ℬ}_{UDD}$ and ${ℬ}_{SDD}$ still carries different local statistics information from the global statistics information of D. In order to ensure the consistency between the local batch data distribution and the global training data distribution, we designed a simple PSR to generate a batch set satisfying the requirement that every example in ℬ is drawn from the global data distribution of D with a replacement.

Differently from the SSR, the PSR firstly draws M labels according to the label distribution of ℬ, where M equals the batch size. Then, for each drawn label, the PSR samples one example randomly from the examples in D whose corresponding labels contain the drawn label. Finally, the sampled M examples constitute a batch. After repeating above procedure $N^{\prime }$ times, the batch set ${ℬ}_{IDD}$ is formed. In this paper, $N^{\prime }$, i.e., the size of the batch set ${ℬ}_{IDD}$, is uniformly set to ⌈|D|/M⌉, too.

## 4. Experiments

### 4.1. Experimental Dataset

In our experiments, the third version of the Medical Information Mart of Intensive Care (MIMIC-III) dataset [33] was adapted for evaluating the effectiveness of our proposed curriculum batching strategy for automatic ICD coding with multi-label classification models. As described in Section 3.1 above, we used one discharge summary (electronic case) of the NOTEEVENTS in MIMIC-III and its corresponding ICD-9 codes to form an example. Next, we conducted a series of data cleaning and preprocessing for all the samples to obtain the experimental dataset D. The specific preprocesses were as follows:All the punctuation, numbers and stop words were removed from all the examples;The discharge summaries were segmented into tokens by using the space as the separator, and then we built a vocabulary $V^{\prime }$ based on these tokens;The TF-IDF values of each word in $V^{\prime }$ are calculated, based on all the examples, and only the top 10,000 words are kept to be used to build the final vocabulary V.

After preprocessing, the basic information of D is listed in Table 2.

As shown in Table 2, D contains 55,177 different electronic medical records with 6918 different diseases in the ICD-9 code. Furthermore, in our experiments, D is further divided by a shuffling algorithm into a training dataset $D_{train}$, a validation dataset $D_{val}$ and a test dataset $D_{test}$ in the ratio of 7:1:2.

### 4.2. Evaluation Metrics

In this paper, widely used micro-measures for multi-label classification tasks, including Precision*_micro_* ($P_{micro}$), Recall*_micro_* ($R_{micro}$), F1-Score*_micro_* ($F1_{micro}$) and F1-Score*_macro_* ($F1_{macro}$), are used for evaluating the model performance, and the equations of these measures are shown as follows:(1)$$P_{micro}=\frac{\mathrm{TP}}{\mathrm{TP}+\mathrm{FP}}$$
(2)$$R_{\mathrm{micro}}=\frac{\mathrm{TP}}{\mathrm{TP}+\mathrm{FN}}$$
(3)$$F1_{micro}=\frac{2\times P_{micro}\times R_{micro}}{P_{micro}+R_{micro}}$$
(4)$$P_{macro}=\frac{1}{\left|L\right|}\sum_{l}^{\left|L\right|}P_{micro}^{l},\;R_{macro}=\frac{1}{\left|L\right|}\sum_{l}^{\left|L\right|}R_{micro}^{l}$$
(5)$$F1_{macro}=\frac{2\times P_{macro}\times R_{macro}}{P_{macro}+R_{macro}}$$
where TP is the number of examples where the predictions are true positives; FP is the number of examples where the predictions are false positives; FN is the number of examples where the predictions are false negatives; and |L| is the size of the label space.

### 4.3. Experimental Settings

In our experiments, we firstly trained TextCNN, TextRNN and TextRCNN for automatic ICD coding on $D_{train}$ and tuned them on $D_{val}$; three models were implemented, based on an open-source tool named NeuralNLP [34]. Finally, we applied the best model on $D_{test}$ to get the results. A shuffling algorithm was applied for generating batches, and the batch size was set to different values to observe the sensitivity of the models. Adam was used as the optimizer, and the learning rate η and the epochs E were set to 0.008 and 150, respectively. The results are listed in Table 3.

CBS was applied on TextCNN, TextRNN and TextRCNN for the automatic ICD coding through directly replacing the original shuffling algorithm-based batching strategy. We refer to them by “TextCNN+CBS”, “TextRNN+CBS” and “TextRCNN+CBS”. ${ℬ}_{UDD}$, ${ℬ}_{SDD}$ and ${ℬ}_{IDD}$ were built offline previously, and they were switched to be utilized in the training procedure every 50 epochs. For the sake of fairness, the other settings are the same as the settings described earlier, and the results are listed in Table 4.

One ReLS method mentioned in [10] was applied directly to TextCNN, TextRNN and TextRCNN as a reference comparison model, and we named the results TextCNN+ReLS, TextRNN+ReLS and TextRCNN+ReLS. The settings of the models are the same as the settings mentioned earlier. Their results are listed in Table 5.

Moreover, every experiment described above was run three times, and the average of the results is reported.

### 4.4. Overall Results

First of all, comparing $F1_{micro}$ in Table 3 with that in Table 4 shows clearly that, after changing the original batching strategy to CBS, the performance of TextCNN, TextRNN and TextRCNN improved by more than double. In particular, the $F1_{macro}$ increased from less than 5% to around 50% ($F1_{macro}$ is more affected by the long-tailed label). Moreover, the results shown in Table 4 and Table 5 demonstrate that, with a simple and effective curriculum batching strategy substitution, the results of TextCNN+CBS, TextRNN+CBS and TextRCNN+CBS are much better than TextCNN+ReLS, TextRNN+ReLS and TextRCNN+ReLS, which use a complex label set reconstruction method. These results demonstrate that our proposed curriculum batching strategy for automatic ICD coding with multi-label classification models is effective.

All the results state that feeding the training data to the deep multi-label ICD coding model with an easy-to-learn data distribution first, and then, gradually, with a hard-to-learn data distribution can effectively improve the model’s performance.

In order to illustrate the ability of our proposed strategy for automatic ICD coding with deep multi-label classification models without harming the generalization ability of the models, we also applied the models TextCNN+CBS, TextRNN+CBS and TextRCNN+CBS directly on $D_{train}$. The results are listed in Table 6. By comparing the $F1_{micro}$ in Table 4 and Table 6, it is easy to find that the performance of the models on the training data is comparable to that on the test data. It means that TextCNN+CBS, TextRNN-+CBS and TextRCNN+CBS do not suffer from the overfitting problem.

### 4.5. Detailed Analysis

It can be seen from the results listed in Table 3, Table 4 and Table 5 that the proposed curriculum batching strategy for automatic ICD coding with deep multi-label classification models improves the results of $F1_{micro}$ as well as the results of $P_{micro}$ and $R_{micro}$. This result is different from the results obtained by the models TextCNN, TextRNN, TextRCNN, TextCNN+ReLS, TextRNN+ReLS and TextRCNN+ReLS. They improve the $P_{micro}$ performance while sacrificing their $R_{micro}$ performance. This further demonstrates that, after applying our proposed CBS to the training procedure, the generalization ability of the trained models has improved.

Our results are also consistent with the results of other existing research. Larger batch sizes result in a lower frequency of gradient updates and better gradient update approximation, but they hurt the performance (as shown in Table 3, Table 4 and Table 5). Therefore, the batch size should not be too large, and, of course, it should also not be too small. In our experiments, when the batch size is set to 1000, the results are the best for TextCNN+CBS and TextRCNN+CBS, and when it is set to 500, the results are the best for TextRNN+CBS.

Moreover, we plotted the training procedures for the nine models whose $F1_{micro}$ are shown in bold in Table 3, Table 4 and Table 5 in order to observe the convergence of every model. The training procedures are grouped according to their base models, i.e., TextCNN, TextRNN and TextRCNN, and presented in Figure 5, Figure 6 and Figure 7, respectively. In the three figures, the X-axis is the number of epochs and the Y-axis is the loss value; the loss is calculated using BCEWITHLOGITSLOSS in PyTroch, and its calculation formula is as follows:(6)ℓ(x,y)=[y·logσ(x)+(1−y)·log (1−σ(x))],

It can be seen in these figures that all the models converge after 150 epochs.

It can also be observed from Figure 5, Figure 6 and Figure 7 that TextCNN+CBS, TextRNN+CBS and TextRCNN+CBS converge the fastest, respectively, in their group. TextCNN+ReLS, TextRNN+ReLS and TextRCNN-+ReLS have the slowest rate of convergence, but they have better $F1_{micro}$ than their base models, TextCNN, TextRNN and TextRCNN. The results show that our proposed curriculum batching strategy for automatic ICD coding with multi-label classification models can help the models to find their better local optima quickly through a gradual learning process without hurting their generalization ability.

Moreover, the loss values of TextCNN+CBS, TextRNN-+CBS and TextRCNN+CBS, in Figure 5, Figure 6 and Figure 7 at the 50th epoch and the 100th epoch, all have two interesting change points showing a steep rise and then a steep drop. These change points occur when ${ℬ}_{UDD}$ switches to ${ℬ}_{SDD}$, and ${ℬ}_{SDD}$ switches to ${ℬ}_{IDD}$, respectively. That means when the local data distribution changes dramatically, the loss will be affected. In other words, the performance of the models would be hurt at those points. However, owing to our proposed curriculum batching strategy, the models can quickly adapt to the new data distributions. It not only keeps good prediction performance, but also avoids the risk of falling into a local optimal solution.

### 4.6. Evaluating the Learning Difficulties of ${ℬ}_{UDD}$, ${ℬ}_{SDD}$ and ${ℬ}_{IDD}$

In this paper, the proposed strategy is based on an intuitive assumption that ${ℬ}_{UDD}$ is easier to be learned by the deep multi-label ICD coding models than ${ℬ}_{SDD}$, and ${ℬ}_{SDD}$ is easier to be learned than ${ℬ}_{IDD}$. The implication of this assumption is that the data distribution of ${ℬ}_{IDD}$ is more like the data distribution of D than that of ${ℬ}_{SDD}$, and the data distribution of ${ℬ}_{SDD}$ is more like that of D than that of ${ℬ}_{UDD}$.

In order to further verify the effectiveness of the proposed curriculum batching strategy, the data distribution similarities between the three generated batch sets and D, i.e., the learning difficulties of ${ℬ}_{UDD}$, ${ℬ}_{SDD}$ and ${ℬ}_{IDD}$, are evaluated by using the Kullback–Leibler Divergence (KLD) and the Jensen–Shannon Distance (JSD). The evaluation results are listed in Table 7. It can clearly see that the data distribution of ${ℬ}_{UDD}$ is the least like that of D. ${ℬ}_{SDD}$ is the next, and the data distribution of ${ℬ}_{IDD}$ is the most like that of D.

### 4.7. Ablation Experiments of CBS

To prove the effectiveness of the curriculum in the CBS, we designed and implemented ablation experiments for the three models, and the experimental results are shown in Table 8. It can be clearly observed that all the performance indicators of the three models are steadily improving with the increase in the curricula.

### 4.8. Evaluating the Long-Tailed Label Learning Performance

The improvements in the automatic ICD coding performance benefit from our proposed curriculum batching strategy. The improvements are also related to the ability of the proposed strategy to promote the learning of the long-tailed labels (labels with a low frequency in D, as shown in Figure 8). In the early phase of the training, we used ${ℬ}_{UDD}$**,** which satisfies the uniform distribution, to train the models. This not only provided a simple learning task, but also allowed the information in the long-tailed labels in the MIMIC-III dataset to be learned fully. To evaluate this, we set a threshold γ = 219. If the frequencies of labels were lower than γ, we considered these labels the long-tailed labels. Then, we calculated the precisions and recalls of the long-tailed labels of the models, and the results are listed in Table 9. It vividly shows that the proposed curriculum batching strategy improves the long-tailed label learning ability of deep multi-label classification models for automatic ICD coding.

## 5. Conclusions and Future Work

In this paper, we observed that extremely large ICD coding label sets and imbalanced label distribution lead to inconsistency between the local batch data distribution and the global training data distribution, and this inconsistency brings the overfitting problem into the mini-batch gradient descent algorithm-based training procedure for deep multi-label classification models for automatic ICD coding. Therefore, we proposed and investigated a simple and effective curriculum batching strategy to replace the original shuffling algorithm-based batching strategy in the training procedure. The proposed strategy emphasizes that the model training procedure should be gradual, rather than random; the proposed curriculum batching strategy realizes a gradual model training procedure by generating three batch sets offline through applying three specific sampling algorithms. These batch sets satisfy the uniform data distribution, the shuffling data distribution and the original training data distribution, respectively. The learning tasks corresponding to these batch sets range from easy to hard. A series of experiments demonstrate the effectiveness of the proposed batching strategy. After replacing the original shuffling algorithm-based batching strategy with the proposed curriculum batching strategy, the performance of the three investigated deep multi-label classification models for automatic ICD coding all improve dramatically. At the same time, the models avoid the overfitting issue and all show better ability to learn the long-tailed label information. In addition, we also compared the performance with a state-of-the-art label set reconstruction model for automatic ICD coding. The performance of ours is consistently higher than that of the state-of-the-art model. These results further prove the effectiveness of the proposed curriculum batching strategy for automatic ICD coding with multi-label classify-cation models.

Possible directions for future work may include choosing the course of the CBS strategy adaptively according to the convergence of the multi-label classification model during the training. Currently, the order of courses and the number of times they are used are fixed. Another direction is to combine transfer learning and pre-train the model with data sets in the medical field.

## Figures and Tables

**Figure 1 healthcare-10-02397-f001:**
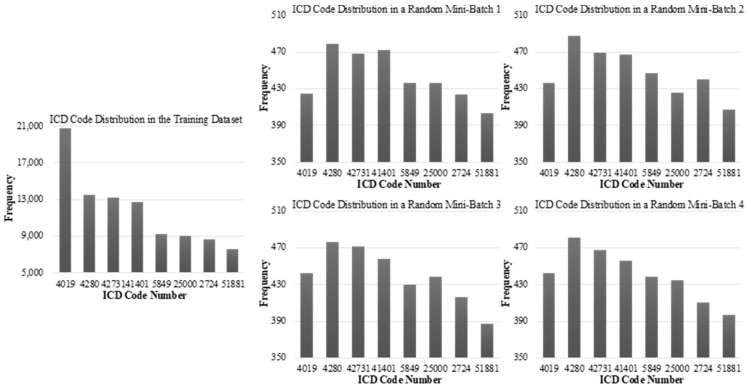
Comparing the global training data distribution (**left**) with sampled local batch data distributions (**right**). The right four charts are drawn based on four batches randomly chosen from one model training procedure. The model trained was based on the original MBGD, and the batch size is 500.

**Figure 2 healthcare-10-02397-f002:**
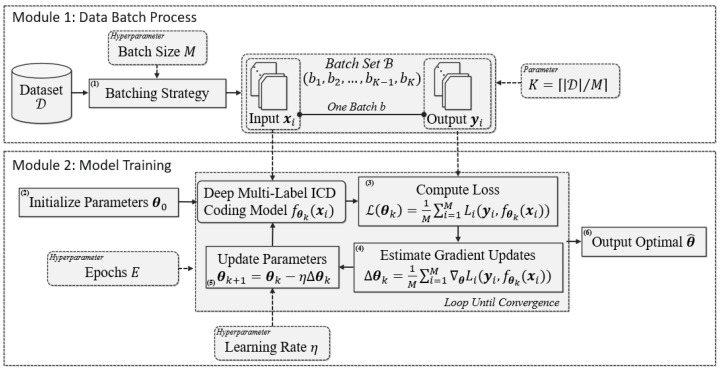
MBGD-based training procedure for the deep multi-label classification models for automatic ICD coding.

**Figure 3 healthcare-10-02397-f003:**
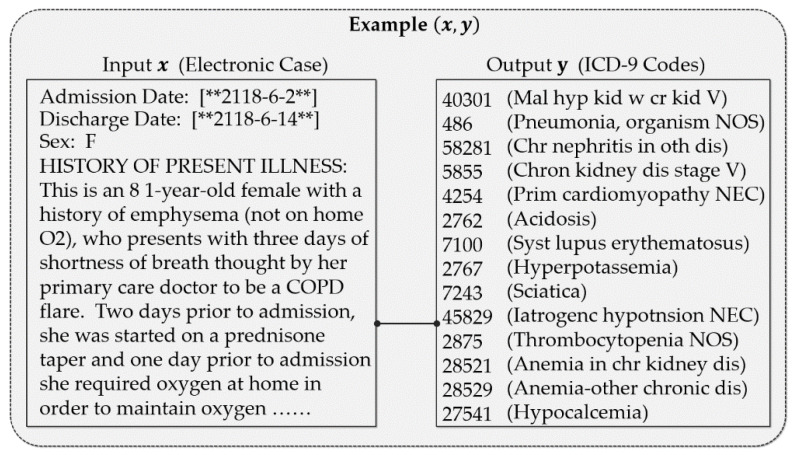
A sample of a real example (x, y).

**Figure 4 healthcare-10-02397-f004:**
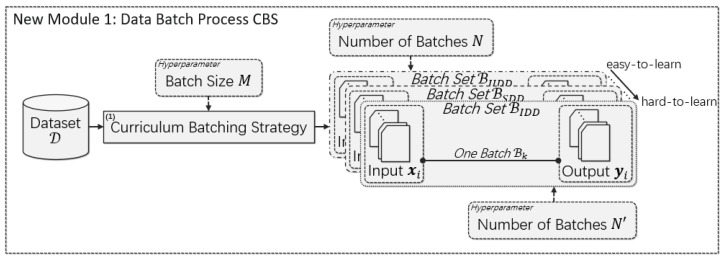
The curriculum batching strategy proposed in this paper has a similar structure to the batching strategy in the original MBGD-based training procedure for deep multi-label classification models for automatic ICD coding. Differently, ${ℬ}_{UDD}$, ${ℬ}_{SDD}$ and ${ℬ}_{IDD}$ will be iterated in turn from easy to hard.

**Figure 5 healthcare-10-02397-f005:**
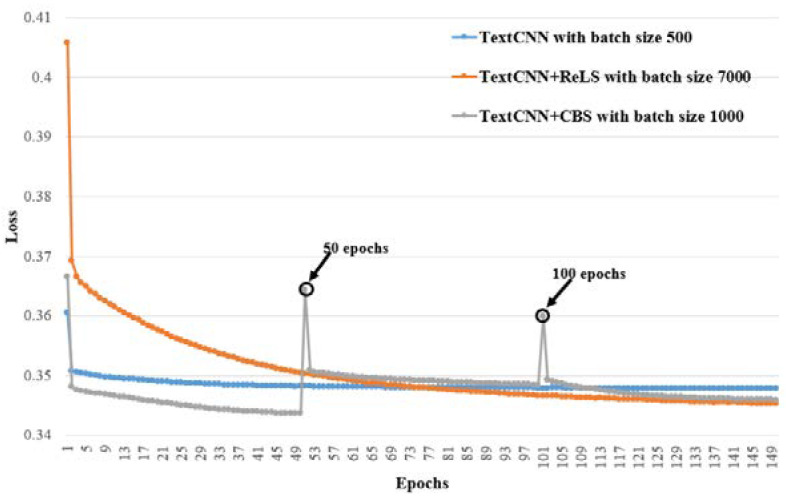
Training procedures for TextCNN-based models with different batch sizes whose $F1_{micro}$ are in bold in Table 3, Table 4 and Table 5.

**Figure 6 healthcare-10-02397-f006:**
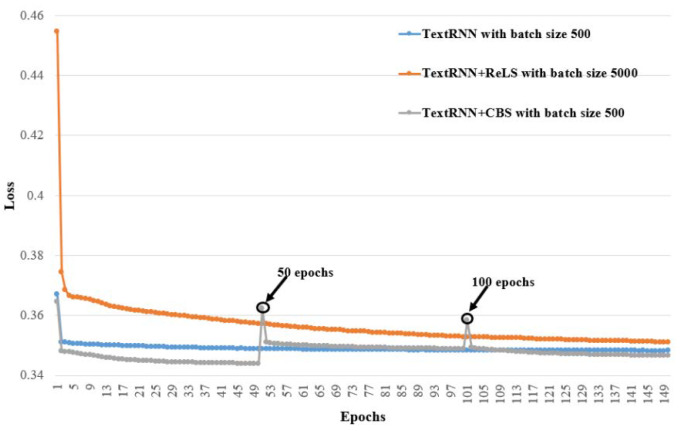
Training procedures for TextRNN-based models with different batch sizes whose $F1_{micro}$ are in bold in Table 3, Table 4 and Table 5.

**Figure 7 healthcare-10-02397-f007:**
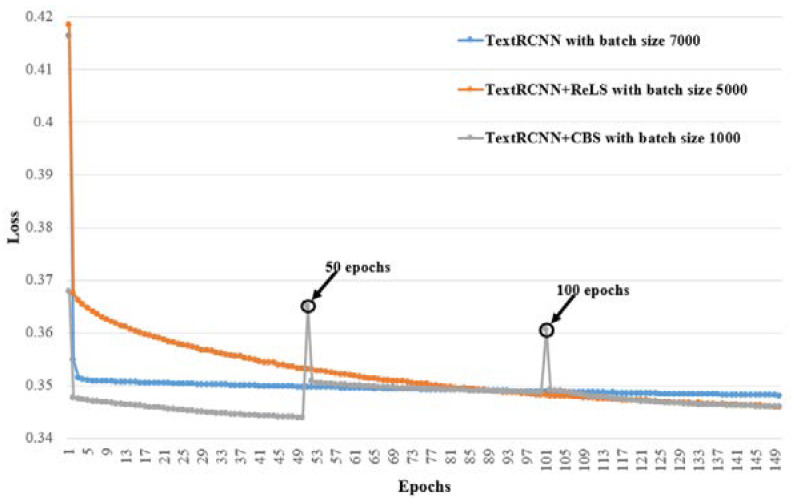
Training procedures for TextRCNN-based models with different batch sizes whose $F1_{micro}$ are in bold in Table 3, Table 4 and Table 5.

**Figure 8 healthcare-10-02397-f008:**
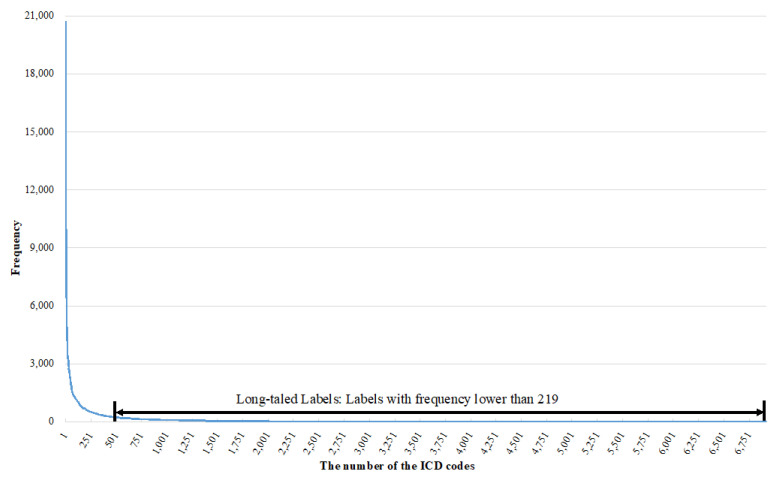
Long-tailed labels in the MIMIC-III dataset and our definition in this paper.

**Table 1 healthcare-10-02397-t001:** Three representative deep multi-label classification models, TextCNN, TextRNN and TextRCNN for automatic ICD coding; all show signs of the overfitting issue. The algorithms obtained decent performance on the training data (top) but observed a large performance gap between training data and test data.

Models	TextCNN	TextRNN	TextRCNN
$F1_{micro}$ on Training Data	0.695	0.789	0.624
$F1_{micro}$ on Test Data	0.325	0.282	0.317

**Table 2 healthcare-10-02397-t002:** The basic information of the MIMIC-III dataset after preprocessing in this paper.

|D|	Unique Labels	Avg. Words per Example	Max Words in Examples
55,177	6918	898	4604
**Min Words** **in Examples**	**Avg. Labels** **per Example**	**Max Labels** **per Example**	**Min Labels** **per Example**
2	11	39	1

**Table 3 healthcare-10-02397-t003:** The automatic ICD coding performance of TextCNN, TextRNN and TextRCNN with different batch sizes. The best results are in bold.

Batch Size	TextCNN				TextRNN				TextRCNN			
	$P_{micro}$	$R_{micro}$	$F1_{micro}$	$F1_{macro}$	$P_{micro}$	$R_{micro}$	$F1_{micro}$	$F1_{macro}$	$P_{micro}$	$R_{micro}$	$F1_{micro}$	$F1_{macro}$
500	0.5365	0.2341	0.3254	0.0237	0.4725	0.2005	0.2815	0.0243	0.4867	0.2324	0.3146	0.0302
1000	0.5456	0.2311	0.3246	0.0222	0.5084	0.1814	0.2674	0.0192	0.5021	0.2265	0.3121	0.0264
2000	0.5397	0.2260	0.3186	0.0216	0.4769	0.1761	0.2572	0.0183	0.4945	0.2250	0.3092	0.0261
5000	0.5578	0.2207	0.3163	0.0197	0.4941	0.1644	0.2467	0.0162	0.4677	0.2379	0.3154	0.0265
7000	0.5403	0.2241	0.3168	0.0214	0.5541	0.1383	0.2214	0.0108	0.4694	0.2401	0.3174	0.0256

**Table 4 healthcare-10-02397-t004:** The automatic ICD coding performance of TextCNN+CBS, TextRNN+CBS and TextRCNN+CBS with different batch sizes. The best results are in bold.

Batch Size	TextCNN+CBS				TextRNN+CBS				TextRCNN+CBS			
	$P_{micro}$	$R_{micro}$	$F1_{micro}$	$F1_{macro}$	$P_{micro}$	$R_{micro}$	$F1_{micro}$	$F1_{macro}$	$P_{micro}$	$R_{micro}$	$F1_{micro}$	$F1_{macro}$
500	0.8121	0.6251	0.7064	0.5789	0.8072	0.4487	0.5766	0.4619	0.8246	0.6004	0.6949	0.5669
1000	0.8403	0.6693	0.7449	0.5802	0.7919	0.3605	0.4954	0.3300	0.8546	0.6167	0.7161	0.5563
2000	0.8507	0.5813	0.6907	0.4234	0.7226	0.2042	0.3184	0.0551	0.8304	0.5230	0.6418	0.4738
5000	0.8432	0.4940	0.6230	0.2898	0.6983	0.1510	0.2483	0.0198	0.8350	0.4142	0.5530	0.3602
7000	0.8005	0.3468	0.4839	0.0882	0.6638	0.1063	0.1832	0.0062	0.7800	0.3240	0.4580	0.1571

**Table 5 healthcare-10-02397-t005:** The automatic ICD coding performance of TextCNN+ReLS, TextRNN+ReLS and TextRCNN+ReLS with different batch sizes. The best results are in bold.

Batch Size	TextCNN+ReLS				TextRNN+ReLS				TextRCNN+ReLS			
	$P_{micro}$	$R_{micro}$	$F1_{micro}$	$F1_{macro}$	$P_{micro}$	$R_{micro}$	$F1_{micro}$	$F1_{macro}$	$P_{micro}$	$R_{micro}$	$F1_{micro}$	$F1_{macro}$
500	0.4987	0.3412	0.4052	0.0977	0.4391	0.3107	0.3639	0.0816	0.4611	0.3471	0.3961	0.1030
1000	0.5091	0.3320	0.4019	0.0973	0.4171	0.3016	0.3501	0.0781	0.5065	0.3401	0.4069	0.0981
2000	0.5574	0.3474	0.4280	0.1010	0.4757	0.3052	0.3719	0.0817	0.5318	0.3624	0.4310	0.1067
5000	0.6290	0.3241	0.4277	0.0829	0.4882	0.3052	0.3756	0.0854	0.5638	0.3529	0.4341	0.1046
7000	0.6190	0.3310	0.4313	0.0824	0.5157	0.2890	0.3704	0.0813	0.5408	0.3567	0.4299	0.1068

**Table 6 healthcare-10-02397-t006:** The automatic ICD coding performance obtained by applying the models TextCNN+CBS, TextRNN+CBS and TextRCNN+CBS directly on the training data.

Batch Size	TextCNN+CBS (On Training Data)				TextRNN+CBS (On Training Data)				TextRCNN+CBS (On Training Data)			
	$P_{micro}$	$R_{micro}$	$F1_{micro}$	$F1_{macro}$	$P_{micro}$	$R_{micro}$	$F1_{micro}$	$F1_{macro}$	$P_{micro}$	$R_{micro}$	$F1_{micro}$	$F1_{macro}$
500	0.7934	0.5898	0.6766	0.8312	0.7886	0.4219	0.5496	0.6854	0.8039	0.5650	0.6636	0.8132
1000	0.8198	0.6353	0.7157	0.8179	0.7755	0.3395	0.4721	0.4451	0.8339	0.5805	0.6842	0.7935
2000	0.8290	0.5472	0.6592	0.5278	0.7195	0.1919	0.3029	0.0612	0.8076	0.4939	0.6130	0.6502
5000	0.8225	0.4658	0.5947	0.3245	0.6984	0.1400	0.2332	0.0208	0.8158	0.3892	0.5263	0.4777
7000	0.7839	0.3269	0.4614	0.0902	0.6672	0.0979	0.1707	0.0066	0.7908	0.2624	0.3938	0.1954

**Table 7 healthcare-10-02397-t007:** Learning difficulties of ${ℬ}_{UDD}$, ${ℬ}_{SDD}$ and ${ℬ}_{IDD}$ evaluated by KLD and JSD, respectively. The smaller the KLD value is, the more difficult to be learned the batch set is. The smaller the JSD value is, the less difficult to be learned the batch set is.

Method	${ℬ}_{UDD}$	${ℬ}_{SDD}$	${ℬ}_{IDD}$
KLD	0.124	0.103	0.090
JSD	33.3 × 10^−3^	12.8 × 10^−3^	8.97 × 10^−3^

**Table 8 healthcare-10-02397-t008:** Results of ablation experiment (batch size = 1000).

Model	Curriculum	$P_{micro}$	$R_{micro}$	$F1_{micro}$	$F1_{macro}$
TextCNN	${ℬ}_{SDD}$	0.5456	0.2311	0.3246	0.0222
	${ℬ}_{UDD}$+${ℬ}_{SDD}$	0.7556	0.3224	0.4407	0.1584
	${ℬ}_{UDD}$+${ℬ}_{SDD}$+${ℬ}_{IDD}$	0.8403	0.6693	0.7449	0.5802
TexRNN	${ℬ}_{SDD}$	0.5084	0.1814	0.2674	0.0192
	${ℬ}_{UDD}$+${ℬ}_{SDD}$	0.7963	0.3832	0.5174	0.2536
	${ℬ}_{UDD}$+${ℬ}_{SDD}$+${ℬ}_{IDD}$	0.7919	0.3605	0.4954	0.3300
TextRCNN	${ℬ}_{SDD}$	0.5021	0.2265	0.3121	0.0264
	${ℬ}_{UDD}$+${ℬ}_{SDD}$	0.7378	0.2590	0.3766	0.1399
	${ℬ}_{UDD}$+${ℬ}_{SDD}$+${ℬ}_{IDD}$	0.8546	0.6167	0.7161	0.5563

**Table 9 healthcare-10-02397-t009:** Comparing the recalls of the long-tailed labels of TextCNN, TextRNN and TextRCNN with their CBS-improved models for automatic ICD coding.

Model	Precision	Recall
TextCNN	0.0188	0.0498
TextRNN	0.0169	0.0442
TextRCNN	0.0103	0.0269
TextCNN+CBS	0.2599	0.6369
TextRNN+CBS	0.2593	0.6617
TextRCNN+CBS	0.1943	0.4778

## Data Availability

The dataset used in this study is from a large-scale freely accessible critical care database—MIMIC-III, which is widely used in ICD coding tasks. MIMIC-III (‘Medical Information Mart for Intensive Care’) is a large, single-center database comprising information relating to patients admitted to critical care units at a large tertiary-care hospital. The data include vital signs, medications, laboratory measurements, observations and notes charted by care providers, fluid balance, procedure codes, diagnostic codes, imaging reports, hospital length of stay, survival data, and more. This dataset is published in the paper by Johnson A E W, Pollard T J, Shen L, et al., *MIMIC-III, a freely accessible critical care database*[J]. MIMIC-III is provided as a collection of commas separated value (CSV) files, along with scripts to help with importing the data into database systems including PostreSQL, MySQL, and MonetDB. Researchers can request access via a process documented on the MIMIC website (http://mimic.physionet.org) (accessed on 16 September 2022). There are two key steps that must be completed before access is granted: the researcher must complete a recognized course in protecting human research participants that includes Health Insurance Portability and Accountability Act (HIPAA) requirements; and the researcher must sign a data use agreement, which outlines appropriate data usage and security standards and forbids efforts to identify individual patients. Approval requires at least a week. Once an application has been approved, the researcher will receive emails containing instructions for downloading the database from PhysioNetWorks, a restricted access component of PhysioNet.

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
