# Peer review of "A Curriculum Batching Strategy for Automatic ICD Coding with Deep Multi-Label Classification Models"

_healthcare, 2022, doi:10.3390/healthcare10122397_

Round 1
Reviewer 1 Report
This paper tried to solve imbalanced label distribution that can bring the problem of inconsistency between the local batch data distribution and the global training data distribution into the minibatch gradient descent (MBGD) using a curriculum batching strategy for improving the MBGD-based training procedure.
There are several comments to improve paper quality.
1. Explanation of methodology is confusing. The paper writing is hard to understand. For example, the authors explain the MBGD-Based Training Procedure and Curriculum Batching Strategy which should be explained by narrative. So that, the reader can understand well the flow of figures 2 and 3. Please change the writing style for easy to understand. It also happened on the Experimental Dataset part.
2. "to randomly arrange examples (??, ??)", what i stands for? "The number of batches ℬ? in ℬ", what k stands for? please explain all variables in the equation. The reason why the explanation is questionable is associated with the first comment.
3. Figures 4, 5, 6, what is the meaning of orange, grey, and blue colors?
Author Response
请参阅附件。

Reviewer 2 Report
The work is devoted to the problems of multi-label classification of diseases based on the standard ICD. The paper is technically sound, the methods used are adequate for the problem. The work is accompanied by the corresponding numerical examples. The main comments are concerning the description of the methodology and presentation of the work's numerical examples. Comments:
1. What is the main question addressed by the research? Please explain the main objective clearly in the abstract and in the introductory section.
2. For the purpose of better presentation, please describe clearly the automatic ICD coding in the introductory section using mathematical terms.
3. Please include some basic info on ICD labels. What is the structure of the label etc? Please explain the usage of real ICD labels with your examples.
4. I consider the topic of the work as original and relevant to the journal. But please describe the field of your study (machine learning or electronic medical records, etc.), where your work
addresses a specific gap in the field.
5. Please explain what your work adds to the subject area compared with other published
material?
6. I think that when describing the methodology, you should consider the improvements regarding the mathematical terms (using the names of the variables as much as you can). You should relate your methodology with the particularities of real ICD.
7. You should explain your experimental study better. What diseases were considered?
8. Please improve your conclusion by adding some discussion and open problems.
Round 2
Reviewer 1 Report
Thank you for the response. All comments from reviewer have been addressed in current version.
Reviewer 2 Report
All my comments were addressed in the revised version. I recommend the manuscript for publication